# Concurrent Infection of the Human Brain with Multiple *Borrelia* Species

**DOI:** 10.3390/ijms242316906

**Published:** 2023-11-29

**Authors:** Maryna Golovchenko, Jakub Opelka, Marie Vancova, Hana Sehadova, Veronika Kralikova, Martin Dobias, Milan Raska, Michal Krupka, Kristyna Sloupenska, Natalie Rudenko

**Affiliations:** 1Biology Centre Czech Academy of Sciences, Institute of Parasitology, 37005 Ceske Budejovice, Czech Republic; vancova@paru.cas.cz; 2Biology Centre Czech Academy of Sciences, Institute of Entomology, 37005 Ceske Budejovice, Czech Republic; j.opelka@email.cz (J.O.); sehadova@yahoo.com (H.S.); 3Faculty of Sciences, University of South Bohemia, 37005 Ceske Budejovice, Czech Republic; 4Institute of Forensic Medicine and Medical Law, University Hospital Olomouc, 77900 Olomouc, Czech Republic; veronika.kralikova@fnol.cz (V.K.); martin.dobias@fnol.cz (M.D.); 5Department of Immunology, University Hospital Olomouc, 77900 Olomouc, Czech Republic; milan.raska@upol.cz; 6Department of Immunology, Faculty of Medicine and Dentistry, Palacky University Olomouc, 77900 Olomouc, Czech Republic; krupka.olomouc@gmail.com (M.K.); sloupenska.k@gmail.com (K.S.)

**Keywords:** *Borrelia burgdorferi*, *Borrelia garinii*, co-infection, human brain, immunohistochemistry, Lyme disease, molecular detection, neuroborreliosis, persistence

## Abstract

Lyme disease (LD) spirochetes are well known to be able to disseminate into the tissues of infected hosts, including humans. The diverse strategies used by spirochetes to avoid the host immune system and persist in the host include active immune suppression, induction of immune tolerance, phase and antigenic variation, intracellular seclusion, changing of morphological and physiological state in varying environments, formation of biofilms and persistent forms, and, importantly, incursion into immune-privileged sites such as the brain. Invasion of immune-privileged sites allows the spirochetes to not only escape from the host immune system but can also reduce the efficacy of antibiotic therapy. Here we present a case of the detection of spirochetal DNA in multiple loci in a LD patient’s post-mortem brain. The presence of co-infection with *Borrelia burgdorferi* sensu stricto and *Borrelia garinii* in this LD patient’s brain was confirmed by PCR. Even though both spirochete species were simultaneously present in human brain tissue, the brain regions where the two species were detected were different and non-overlapping. The presence of atypical spirochete morphology was noted by immunohistochemistry of the brain samples. Atypical morphology was also found in the tissues of experimentally infected mice, which were used as a control.

## 1. Introduction

Lyme disease (LD) or Lyme borreliosis (LB) is a multi-system disorder with a diverse spectrum of clinical manifestations. It is caused by the spirochetes of the *Borrelia burgdorferi* sensu lato (s.l.) complex. The initial stage of infection is characterized by flu-like symptoms (malaise, fatigue, headache, arthralgias, myalgias, fever, and regional lymphadenopathy) and/or skin rash due to *Borrelia afzelii* (acrodermatitis chronica atrophicans; ACA) and/or various manifestations of erythema migrans (EM) developing within a few weeks after the tick bites. If the spirochete is not eliminated, it will further disseminate to secondary sites of infection, leading to persistent inflammatory pathology, particularly affecting the central nervous system (CNS), peripheral nervous system (PNS), joints, or heart [1]. Symptoms of the secondary stage of infection vary and may disappear after days to months or continue as the disease transitions to the late stage of persistent disease with various signs and symptoms, including fatigue, headaches, sleep disturbance, neuropsychiatric and cognitive impairments, and arthralgias, myalgias, and neuropathy [2,3,4]. Even though antibiotic treatments for the majority of infected patients result in a full recovery, some patients suffer from long-lasting neurological and psychological manifestations. Antibiotic therapy prescribed during late-stage disease can result in unpredictable responses with ongoing symptomatology [5,6,7,8].

Those patients who received standard antibiotic treatment with problems persisting more than six months after antibiotic treatment are sometimes referred to as Post-Treatment Lyme Disease Syndrome (PTLDS) [9]. The etiology of this syndrome is debated, but several hypotheses have been proposed [10]. Persistence of *Borrelia* or associated co-infections, immune dysregulation leading to inflammation or autoimmunity, and/or disrupted central neural pathways [11] leading to central sensitization, among others, have been postulated. If PTLDS is due to or is independent of microbial persistence or other causes remains a topic of debate [12,13]. However, the ability of LD spirochetes to colonize multiple host tissues has been confirmed [14]. Dermis is the first tissue that spirochetes colonize after the tick bite. At this point, the host immune system can control the pathogen burden in the tissue [15]. Colonization of distant tissues involves spirochete dissemination from the dermis, which is mediated by differential regulation of virulence determinants of *B. burgdorferi* s.l. [16] that support the migration of pathogens through the endothelial and blood-brain barriers [17]. Once established in immune-privileged sites (fibroblasts, endothelial cells, macrophages, or neuronal glial cells) [18,19,20,21], the pathogen is capable of triggering a local inflammatory response but is safe from being cleared by the host immune system or antibiotics, which cannot easily penetrate the blood-brain barrier [22]. Survival of spirochetes despite antibiotic treatments, leading to the establishment of chronic LD, has been clearly demonstrated in animal studies [5,23,24,25,26]. 

Multiple species from the *B. burgdorferi* s.l. complex are responsible for LD, particularly in Europe [27]. The main burden of human cases, approximately 60%, is linked to infection with *B. afzelii*, which leads to the most common manifestation of skin lesions [28]. Three types of skin manifestations of LD are recognized: at stage 1, it is an erythema migrans with a characteristic “bull’s eye rash,” which, if untreated, can be followed by early disseminated infection, borrelial lymphocytoma or multiple EM (stage 2) along with neurologic and cardiac abnormalities, and late infection, especially arthritis in North America or acrodermatitis chronica atrophicans (ACA) (stage 3) in Europe [29].

The second, most represented European *Borrelia* genospecies, *Borrelia garinii*, causes primarily Lyme neuroborreliosis (LNB), affecting both the central and peripheral nervous systems [30]; up to 15% of LD patients suffer from LNB [22,30]. Intracellular localization of LD spirochetes in neurons and glial cells has been confirmed both in vitro and in vivo [12,31,32]. Evidence of *Borrelia* persistence in the brains of chronic LNB patients is limited but has been confirmed [33,34]. Further, the development of dementia, cortical atrophy, or amyloid deposition in some cases has been confirmed as well [12,31]. Invading neurons and glial cells, LD spirochetes can trigger progressive cell death or cause cell dysfunction [35], and the development of dementia, cortical atrophy, or amyloid deposition in some cases has been confirmed as well [12,31]. *Borrelia burgdorferi* sensu stricto (s.s.) is the major cause of LD in North America; however, its impact on European LD is under-appreciated [36,37]. The most frequent manifestation of *B. burgdorferi* s.s.-induced neuroborreliosis in the United States is lymphocytic meningitis, whereas European *B. garinii*-induced LNB, in the majority of cases, is diagnosed as Bannwarth syndrome, with the classic triad of meningitis, cranial neuritis, and painful radiculoneuritis [38], an uncommon manifestation of LNB in the USA [39]. Such differences in clinical manifestations of LNB might be based on different mechanisms of dissemination of the bacterial pathogen into the nervous system, different capabilities of individual species to cross the blood-brain barrier (BBB) by either transcellular or paracellular penetration [40], or hypervariable OspC genotypes [41]. Until recently, no reliable system for the detection of persistent infections existed. A single study on non-human primates showed that persistent forms of spirochetes that survived antibiotic treatment remain metabolically active [42]. A xenodiagnostic study using the laboratory-reared larvae of *Ixodes scapularis* [43] showed that in a group of 10 patients who had high levels of anti-*Borrelia* antibodies after antibiotic treatment, one patient with PLTDS and a patient with EM had positive results, confirming the presence of viable *Borrelia*; two others from the same study group had “indeterminate results.” Further, up to 14.5% of long-term Lyme patients were proven to be PCR positive after seemingly adequate antibiotic (ATB) therapy in a study by Horowitz and colleagues [10]. Immunohistochemistry (IHC) combined with confocal microscopy found significant pathological changes, including borrelial spirochetal clusters, in the autopsy tissues of all the organs of a LD patient who had received extensive antibiotic treatments during her 16-year-long course of the disease [34]. Additional evidence of persistent *Borrelia* infection in patients with ongoing LD symptoms despite extensive ATB therapy has come from the cultivation of *Borrelia* in the blood of seven subjects, from the genital secretions of ten subjects, and from a skin lesion in one subject out of 12 selected LD patients [44]. 

Here we report the case of a patient who, after being infected by *Borrelia* and treated with antibiotics continuously, progressed toward neurologic/psychiatric symptoms over the subsequent 13 years. After this period, the patient underwent repeated serological testing with borderline positivity for *Borrelia* infection, followed by the prescription of several antibiotics, which provided no clinical improvements, followed by hospitalization at psychiatric clinics. Several months after being released from the clinic, the patient committed suicide, providing written consent for analyzing his brain for the presence of *Borrelia*.

## 2. Results

### 2.1. Patient History and Borrelia Serology

A male patient born in 1996 contacted a physician in 2004 after the appearance of erythema migrans. He was diagnosed with LD and was treated with antibiotics (type and duration of treatment are unavailable). The patient progressed to suffer from neurological symptoms, mostly cognitive deficits such as “brain fog”, reduced psychomotor performance, and difficulties with concentration and processing of visual and auditory stimuli. In 2017, the patient was examined in the neurology department and had a positive *Borrelia* serology test, with both IgG and IgM anti-*Borrelia* antibodies in serum, but a negative PCR from a lumbar puncture. In December 2017, the patient was admitted to the Psychiatric Department of the University Hospital Olomouc. His therapy started with Zyprexa (Olanzapin, Eli Lilly, Indianapolis, IN, USA) and was later replaced with Brintellix (Vortioxetin, H.Lundbeck A/S, Copenhagen, Denmark). On February 2018, anti-*Borrelia* antibodies were determined again in a private medical facility with borderline positivity for *Borrelia*-specific IgM and strong “++” positivity for IgG by ELISA. The ELISA results were confirmed by immunoblotting; the IgM immunoblot was positive for the presence of OspC from *B. afzelii*, *B. garinii*, and *B. burgdorferi* s.s., and the IgG immunoblot was borderline positive for VlsE from *B. afzelii*, *B. garinii*, and *B. burgdorferi* s.s., and negative for p83, flagellin, BmpA, OspA, OspB, OspC, and DbpA. Serological tests for *Chlamydia* sp., *Mycoplasma* sp., *Anaplasma phagocytophillum*, *Toxocara canis*, and *Toxoplasma gondii* were negative. Serological testing for *Bartonella* infection by indirect immunofluorescence assay was negative in the IgM and IgG classes for both *Bartonella henselae* and *Bartonella quintana*. A PCR test for the presence of *Babesia* spp. (*Babesia microti* and *Babesia divergence*) in peripheral blood was negative. At the same private medical facility, the patient was prescribed a combination of antimicrobials: 3 × 100 mg/day of minocycline (Ratiopharm, Ulm, Germany), 250 mg (3×/week) of azithromycin (Sandoz, Basel, Switzerland), and 200 mg (1×/day) of hydroxychloroquine (Plaquenil) (Sanofi, Paris, France). In August 2018, the suspected hypocorticism (adrenal insufficiency) was ruled out by biochemical laboratory examinations. In September 2018, the patient was hospitalized at the Department of Psychiatry, University Hospital Olomouc, with the suspicion that he was developing a mental disorder. The patient was diagnosed with schizotypal personality disorder (STPD) and somatoform disorder and was discharged three weeks later. In August 2019, the patient committed suicide. The patient left behind a letter expressing the urgent demand for scientists to analyze his brain for the presence of LD spirochetes. The letter was provided to the Ethical committee of University Hospital Olomouc and initiated this study. The patient’s body was autopsied in the Institute of Forensic Medicine and Medical Law, University Hospital Olomouc, 2 days after his death. 

### 2.2. Post-Mortem Toxicology and Microbiology

Toxicological post-mortem analysis confirmed the presence of the following chemical agents in the blood: hydroxychloroquine, hydroxyzine (the active substance in the anxiolytic Atarax), and its metabolite cetirizine. In addition, a low concentration of azithromycin was detected in cerebrospinal fluid (CSF). Immunoblot analyses of post-mortem serum and CSF conducted at the Faculty of Medicine and Dentistry, Palacky University Olomouc, confirmed borderline IgG reactivity against VlsE of *B. garinii* and *B. burgdorferi* s.s. The reactions with VlsE antigens of *B. afzelii*, lipids of *B. garinii* and *B. afzelii*, p83, p41, p39, OspC, p58, p21, p20, p19, and p18 were negative. Using immunoblot kits for autoantibody detection, weak IgG positivity against PL-7 antigen (threonyl-tRNA synthetase) and borderline reactions with SRP antigen and histones were detected. As a part of the microbiological analysis of post-mortem samples, the CSF and brain tissues were used for the cultivation of spirochetes in the BSK-H medium. After 2 months of incubation of seeded cultures at 34 °C, the presence of live bacteria was not detected in any culture, and all samples were deemed to be culture-negative. 

### 2.3. PCR Detection of Borrelia DNA in Frozen Brain Tissue Samples

To determine if *Borrelia* DNA could be detected in the brain, PCR analysis was performed on DNA from seven different parts of the brain, samples of which had been frozen at −80 °C upon collection. Nine genes (*flaB*, *ospC*, *clpA*, *clpX*, *nifS*, *rplB*, *pepX*, *pyrG*, and *uvrA*) provided the amplicons of the expected size (Appendix A), although not all amplifications were successful for all tested brain loci (Table 1). 

Sequence analysis, followed by comparison with available databases, confirmed the presence of DNA in two spirochete species, *B. burgdorferi* s.s. and *B. garinii*. The two *Borrelia* genospecies were found in brain samples and were found to be mutually exclusive: the DNA of a single species, *B. burgdorferi* s.s., was detected in the temporal right lobe, choroid plexus (left), occipital lobe (left), frontal lobe (left), and parietal lobe (right), while the DNA of *B. garinii* was identified in the basal ganglia (right) and cerebellum (right). No PCR amplification showed the presence of more than one spirochete species in any sample. 

*BLASTn* analysis of the *ospC* sequences confirmed that the *B. burgdorferi* s.s. strain carried an *ospC* of type A, the globally most common *ospC* type. The *B. garinii ospC* sequences detected in the brain samples were identical to strains widely distributed in Eurasia.

### 2.4. Occipital Lobe Tissue Exhibited Structures Consistent with Borrelia

Immunohistochemical (IHC) investigation of paraffin sections of the patient occipital lobe samples (Figure 1) revealed the presence of structures reactive with *Borrelia*-specific rabbit polyclonal antibodies (Figure 2 and Figure 3). Polyclonal, rather than monoclonal, antibodies were selected as they react with a broader spectrum of *Borrelia* antigens, and postmortem degradation might limit the number of borrelial antigens.

The size and morphology of these structures resembled the *Borrelia* cells with the atypical morphology detected previously as well from cultured spirochetes treated with doxycycline or amoxicillin, used as an in vitro test of antibody specificity (see Appendix A and Methods and Appendix A). In *Borrelia* populations treated by both low (50 μg/mL) and high (100 μg/mL) concentrations of either doxycycline or amoxicillin, these antibodies stained both spiral and atypical *B. burgdorferi* forms (Appendix A).

The presence of multiple forms of *Borrelia* in the culture was verified using transmission electron microscopy (TEM) (Appendix A). The structures detected in the samples of the patient’s occipital lobe had a diameter of about 1–10 μm, and presumed protoplasmic cylinders ranged between 0.2 and 0.4 µm in diameter. These structures were detected with an estimated frequency of 0.16–0.3 per 1 mm^3^ of tissue typically located near the capillaries (Figure 2A,B). *Borrelia* cyst-like structures were also detected in the paraplast section of a sample from the patient’s occipital lobe (Figure 3). Anti-*B.burgdorferi* polyclonal antibodies visualized by the horseradish peroxidase-conjugated secondary antibodies were used for the detection. Our findings on human brain tissues were in accordance with the observation of tissue samples from mice artificially infected with *B. burgdorferi* s.s. and subsequently treated with antibiotics that we used as spiked controls (Figure 4). The structures corresponding to the atypical forms of *Borrelia* in the mouse tissues were mainly found in samples from bladders (Figure 4A,B) and knee joints (Figure 4C,D).

## 3. Discussion

Detection of *Borrelia* in multiple organs of infected animals, including humans, demonstrates the ability of these spirochetes to disseminate into the secondary sites of infection (to review, see [12] (citations 17–49). Detection of intact spirochetes at the sites of secondary infection in both laboratory animals and humans even after aggressive antibiotic treatments further demonstrates the ability of the spirochetes to persist ([22]; to review, see [12] (citations 50–85)). The Centers for Disease Control and Prevention (CDC, USA) indicate that up to 20% of patients continue to suffer from neurological manifestations such as persistent fatigue and joint or muscle pain [45] after seemingly successful antibiotic treatment; however, recent large-scale testing revealed that the real number of antibiotic-treated patients that develop symptoms of chronic disease is between 36 and 63% [46]. The role of biofilm and stationary persister forms of *Borrelia* in persistent LD is not clear; however, such morphologies are an example of bacterial adaptation to a changing environment. As complex structures highly resistant to environmental and therapeutic pressure, biofilms are recognized as an essential part of the mechanism of the establishment of chronic infection [12,47,48,49,50,51,52,53,54,55]. The ability of spirochetes to survive antibiotic treatment to instigate reoccurrence of the disease is strong evidence that chronic LD could arise from persisting and re-occurring infection, either caused by persisting forms hidden in biofilms, cell-wall deficient forms, round bodies, or other alternative spirochete morphologies [12]. These previous findings are consistent with our finding of spirochetes in a human brain despite extended antibiotic treatment. The detection of intact spirochetes in autopsy brain specimens of humans after extended treatments and those with a diverse history of disease manifestations, including widely recognized neurocognitive disorders, anxiety, depression, memory loss, brain atrophy, and progressive dementia [31,56,57,58,59,60], illustrates that persistent *Borrelia* infection can lead to a persistent disorder of the central nervous system (CNS) and the development of LNB [22,61,62]. It is thought that LNB does not reoccur frequently without reinfection. Nevertheless, cases of reoccurrence of LNB might suggest the formation of biofilm-protective structures and may explain the low rate of spirochete detection in blood in patients with persistent infection [63]. The neurotropic nature of LD spirochetes helps ensure their survival in the CNS [64] for an extended period of time in alternative morphological forms [65,66,67], including antibiotic-resistant biofilms [34]. In general, dissemination of spirochetes occurs through the bloodstream, where the spirochetes remain for a short period of time before moving to the extracellular matrix of various internal organs, where they can remain shielded from the host’s immune system and antibiotics [16,68,69,70,71,72,73]. However, LD spirochetes were found to also be present in cardiac myocytes, endothelial cells, and synovial cells [19,74,75]. Dissemination of spirochetes into the CNS is thought to occur via passage along the peripheral nerves [35]. Based on the distinct clinical manifestations of LNB in Europe and North America, ascribed to the different *Borrelia* species, *B. garinii* and *B. burgdorferi* s.s., respectively, the mechanisms of spirochete dissemination into the CNS might be species-dependent and rely on the ability of *Borrelia* to cross the BBB [39,40,76]. Some *Borrelia* genotypes may be able to disseminate through the host more effectively than the others or have an advantage competing with less efficient types due to the *ospC* genotypes they possess [41,77]. An association between *ospC* genotypes and the severity of LD in patients has been reported in multiple studies [41,78,79,80]. To date, out of 35 recognized *ospC* genotypes, four, A, B, I, and K, are responsible for systemic LD in humans around the world [77,78,81], although genotypes C, D, N, F, H, E, G, and M have been found in the secondary sites of infection [81,82,83,84]. While some *Borrelia ospC* genotypes cause the typical erythema migrans and the others disseminate through the bloodstream or CNS, only certain strains may exhibit neurotropism [41].

Our results confirmed the presence of two species of spirochetes from the *B. burgdorferi* sensu lato complex, *B. garinii* and *B. burgdorferi* s.s., in different areas of the human brain. Importantly, the DNAs of the two spirochete species were detected in distinct areas of the brain; in no case did we find infection with both in the same brain region. Multiple repeated PCR amplifications with different sets of primers, followed by sequence confirmation of *Borrelia* spp., always provided clear and definite chromatograms indicating the presence of DNA from only one species. Based on the analysis of a single brain tissue infected with spirochetes, it is not yet possible to conclude whether this finding represents biological restriction in brain colonization, the outcome of separate and rare colonization of the brain, or is just a coincidence. The specific mechanisms by which *Borrelia* crosses the blood-brain barrier are not yet fully understood, and an animal model allowing examination of the full pathogenic process has not yet been established, although a recent model using inbred mice has shown colonization of the dura mater during acute and late spirochete infection [85,86]. In this model, a decrease in the inflammatory and immune pathways was observed in dura mater over time [85]. It seems that the post-treatment stage of LNB is not triggered by either immune or inflammatory response pathways, which requires new mechanistic insight to identify ways of clinical intervention [41]. The development of LNB depends on the ability of spirochetes to cross the BBB. This invasion can occur via breaching either physical (tight junctions) or metabolic (enzymes, transport systems) barriers [87]. The role of the BBB in neurodegenerative and neuropsychiatric disorders is crucial; its failure plays an important role in the pathogenesis of many diseases of the CNS that are caused by bacteria or protozoa [88,89,90]. Recent studies show that, in the case of some neurocognitive disorders that lead to the development of dementia, as well as during normal aging, the “leakage” of the BBB is increased [91,92]. Whether *Borrelia* enter the brain by direct transmigration or are carried across the BBB hidden inside non-phagocyting leukocytes using the “Trojan horse” mechanism as in the case of other pathogens is not yet known [89,90]. It is tempting to speculate that some kind of competition might occur between the spirochete species in the process of crossing the BBB, and this might be the basis of our observations of species-segregated *Borrelia* loci in the brain. A recent study by Adams and colleagues [40] showed that different species of LD spirochetes are able to enter BBB-organoids with different success. The spirochetes that successfully invaded the organoids remained viable inside the BBB-organoids, initiating the loss of tight junctions and changes in the organoids’ gross morphology and integrity [40]. 

Unambiguous attribution of persistent neurological symptoms to LD and *Borrelia* persistence is limited by several factors, including the presence of comorbid disease with overlapping symptoms or reinfection with *Borrelia*. Horowitz has described up to 16 different overlapping factors that can account for persistent symptoms [10,54,93,94]. The multiple results of studies of relapsing fever (RF) in humans and animals confirmed that RF spirochetes cause neurological abnormalities as often as LD spirochetes, infecting the brain and other nervous tissues and persisting in the brain after treatment with antibiotics that do not readily penetrate the blood-brain barrier. Among the wide spectrum of neurological manifestations of tick-borne relapsing fever (i.e., meningismus, facial palsy, encephalitis, myelitis, radiculitis, and deafness), the neuropsychiatric abnormalities in patients manifested by the acute onset of mental confusion or excitement, persistent hallucinations, signs of mania, extreme irritability, flight of ideas, persistent fatigue, or neurasthenia [95]. Possibly, the so-called “residual brain infection” is a common phenomenon for all spirochetal diseases. In experimental animals, spirochete infection persisted in the brain for up to 3 years [96,97,98,99,100]. Among the nervous system tissues in which the spirochetes were detected were the cerebral hemispheres, cerebellum, spinal cord, medulla, and cortex. Interestingly, during the residual brain infection, the animals were protected against re-infection with the same strain [101,102]; spirochetes that cause residual brain infection were susceptible to the serum of the animal from which they were recovered earlier [102,103]. It is obvious that the immune response cannot effectively reach the spirochetes in the brain. The brain remains infectious, while the blood of the animal is not. Inoculation of the brain suspension in susceptible experimental animals results in the development of spirochetemia [104]. The strong neurotropism of spirochetes is widely recognized. Multiple pieces of evidence have confirmed the fact that spirochetes can invade the brain and cause persistent infection [58,105,106]. They are able to escape destruction by the host immune system and initiate a chronic inflammatory process. Evidence of infection of the brain in diseases with clear spirochete etiology, such as Lyme disease, relapsing fever, syphilis, and leptospirosis, for example, shows the strong association between the spirochetes and diseases. In addition, spirochetal etiology in other neurological diseases, such as Alzheimer’s disease (AD), has been postulated [31]. *B. burgdorferi*-specific antigens were detected in the brain of an AD patient with concurrent LNB. These findings were supported by positive PCR detection of *B. burgdorferi* DNA in the brain [107,108]. *B. burgdorferi* was also cultivated from the brain in AD patients by MacDonald and Miranda [109], MacDonald [110], and Miklossy [31]. 

Since the recognition of Lyme disease, thousands of studies have been published, yet optimal therapy is still a matter of debate [111]. The complexity of the pathophysiology of the *Borrelia* pathogen and the clinical uncertainty surrounding the signs and symptoms of Lyme and other tick-borne diseases make the choice of treatment a complex one that relies on the clinical judgment of the treating physician. Recognized persistent bacterial infections may require prolonged antibiotic therapy and seem reasonable and justifiable in some situations when considering patients with persistent LD symptoms [111]. However, recent scientific studies on the biofilm and persister forms of *Borrelia* and associated co-infections, along with early clinical trials, provide hope that shorter-term protocols with anti-persister drug regimens might lead to more effective treatments for those suffering from neuropsychiatric and other symptoms of chronic Lyme disease or post-treatment Lyme disease symptoms (PTLDS) [52,53,54,55,94]. 

To our knowledge, this is the first study that confirms the presence of *B. garinii* and *B. burgdorferi* s.s. structures in the human brain with strict separation of invaded brain loci. The question of what defines the distribution of spirochete species in brain tissues remains unanswered but is one of increasing importance as the number of human *Borrelia* infections increases. The possible pathological effect of persistent forms of *Borrelia* detected in the present study is also discussed from the point of view of the potential induction of an immunopathological reaction associated with chronic inflammation and/or autoreactive immune response, where non-replicative structures can be recognized as long-term antigenic stimuli [13].

## 4. Materials and Methods

### 4.1. Ethical Statement

Samples that were analyzed in this study originate from a young adult male who committed suicide in August 2019. The processing of post-mortem samples was performed based on informed consent provided by him before suicide in the form of a letter, which was accepted by the Ethical Committee of University Hospital Olomouc, 102/18 (NV19-05-00191).

### 4.2. CSF and Blood Collection and Analysis

The autopsy was performed at the Institute of Forensic Medicine and Medical Law University Hospital Olomouc and Faculty of Medicine and Dentistry, Palacky University Olomouc, Czech Republic, two days after the suicide. Peripheral blood and cerebrospinal fluid (CSF) samples were collected in a 2 mL volume. The CSF was markedly stained by the presence of blood. Part of the CSF was aseptically removed for the cultivation of spirochetes (Appendix A and Methods). The remaining volume of CSF was centrifuged (2.000× *g*, 10 min, 4 °C) to remove cells, and the supernatant was collected and stored at −80 °C. The blood was centrifuged (2.000× *g*, 10 min, 4 °C), clarified serum was collected, transferred to a new tube, and stored at −80 °C. Serological tests were performed by standard protocols using the Anti-*Borrelia* EUROLINE-RN-AT, EUROLINE Autoimmune Inflammatory Myopathies 16 Ag (IgG), and EUROLINE ANA Profile 3 plus DSF70 (IgG) (EUROIMMUN, Lübeck, Germany) blot diagnostic kit with evaluation by a flatbed scanner and the software EUROLineScan Software 3.4 (EUROIMMUN, Lübeck, Germany).

### 4.3. Brain Tissue Collection (Post-Mortem)

Brain tissue samples of about 9 cm^3^ from seven different parts of the brain were collected post-mortem by certified pathologists at the Institute of Forensic Medicine and Medical Law, University Hospital Olomouc. Brain samples were collected from: 1—temporal lobe (right); 2—choroid plexus (left); 3—occipital lobe (left); 4—frontal lobe (left); 5—parietal lobe (right); 6—basal ganglia (right); and 7—cerebellum (right). All tissue samples were divided into three parts. One part was used for the cultivation of potentially live spirochetes. This sample was immediately aseptically transferred to BSK-H medium supplemented with 6% rabbit serum and antibiotics. The second part was used for PCR analyses. This portion was frozen at −80 °C until use. The third part was used for immunohistochemical analyses. This part was fixed in 10% buffered formalin (4% paraformaldehyde) and stored at 4 °C. 

Based on PCR results, 5 samples of 125 mm^3^ in size were subsequently taken from the fixed autopsied occipital lobe sample for immunohistochemical staining (the sample position within the occipital lobe autopsy is shown in Figure 1).

### 4.4. Analysis of Total DNA from Human Brain Tissues: Polymerase Chain Reactions (PCR)

The DNA purification steps (Appendix A and Methods), PCR, and post-amplification analyses were all performed in separate areas with all precautions against contamination. The presence of *Borrelia burgdorferi* s. l. DNA in the samples was assessed by PCR amplification of partial genes encoding outer surface protein C (OspC) and flagellin, followed by amplification of eight housekeeping genes according to the previously described MLST protocol [112]. To reduce the inhibition of the reaction from the excess of human DNA in the template DNA, the PCR reactions were conducted as nested PCRs under conditions previously used with human samples [113,114]. To assure the absence of possible contamination during the DNA purification or cross-reactivity with the human brain tissue, negative extraction controls were included. The partial *ospC* and *flagellin* genes were amplified by nested PCR using the previously described primers (Appendix A), and the conditions of the reaction were as follows: 30 cycles of denaturation at 95 °C for 30 sec, annealing at 50 °C and 52 °C for 30 s for the external and internal rounds of PCR, respectively, and extension at 72 °C for 30 s [115,116]. Two-step amplification of eight housekeeping genes included: *clpA* (BB0369), *clpX* (BB0612), *pyrG* (BB0575), *uvrA* (BB0837), *pepX* (BB0627), *recG* (BB0581), *rplB* (BB0481, seminested PCR), and *nifS* (BB0084, seminested PCR) (Appendix A). The PCR conditions for the housekeeping genes, except for ***recG***, were as follows: initial denaturation at 95 °C for 15 min, cyclic denaturation at 94 °C for 30 s, annealing temperature from 55 °C to 48 °C (touchdown PCR, decreasing 1 °C each cycle) for 30 s, and the extension step at 72 °C for 60 s. An additional 20 cycles were run using denaturation at 94 °C for 30 s, annealing at 48 °C for 30 s, and extension at 72 °C for 60 s. After a final extension step at 72 °C for 5 min, the samples were kept at 14 °C until the second (nested) PCR. The conditions for the second PCR were as follows: 95 °C for 7 min, followed by 35 cycles of [denaturation at 94 °C for 30 s, annealing at 50 °C for 30 s, extension at 72 °C for 60 s]. After a final extension step for 5 min at 72 °C, the samples were kept at 14 °C.

For *recG,* the PCR conditions for the first set of cycles consisted of initial denaturation at 95 °C for 15 min, followed by 30 cycles of denaturation at 94 °C for 30 s, annealing at 55 °C for 30 s, extension at 72 °C for 60 s, and final extension at 72 °C for 5 min. The conditions for the second set of cycles were identical for all the primers used.

The PCR reactions were carried out in a final volume of 20 μL using 2× HotStarTaq Plus Master Mix (Qiagen, Hilden, Germany). Amplicons were visualized by electrophoresis in a 1.5% agarose gel (1 × TAE, pH 8.0). In all cases, a reaction mix with water instead of a DNA template was used as the negative control. Borrelia carolinensis DNA was used as a positive control in all PCR reactions.

### 4.5. Immunohistochemical Detection of Borrelia in the Paraplast Section of a Human Brain Autopsy and Infected Mouse Tissues

Mice were infected with *B. burgdorferi* s.s. administered simultaneously by intradermal and intraperitoneal routes (Appendix A and Methods). The fixed, autopsied human occipital lobe samples were washed several times in phosphate-buffered saline (PBS). Organs dissected from euthanized mice were submerged in a fixative of saturated picric acid, 4% formaldehyde, and 2.3% copper acetate supplemented with mercuric chloride (Bouin-Hollande solution) [117] overnight at 4 °C. The fixative was then thoroughly washed with 70% ethanol. Standard techniques were used for both human autopsies and mouse tissue samples, including dehydration, embedding in paraplast, sectioning to 10 μm, deparaffinization, and rehydration. The sections were treated with Lugol’s iodine, followed by a 7.5% solution of sodium thiosulphate to remove residuals of heavy metal ions, and then washed in distilled water and PBS supplemented with 0.3% Tween 20 (PBS-Tw). The nonspecific binding sites were blocked with 5% normal goat serum in PBS-Tw (blocking solution) for 30 min at room temperature (RT). Incubation with rabbit polyclonal *B. burgdorferi* antibodies specific to a pool of *B. burgdorferi* s.l. complex proteins (cat.# PA1-73004, Invitrogen, Carlsbad, CA, USA) diluted at 1:200 in the blocking solution was carried out in a humidified chamber overnight at 4 °C, followed by washing the samples by thorough rinsing with PBS-Tw (three times for 10 min at RT) [118,119]. For enzymatic staining, the sections were further incubated in the cross-adsorbed horseradish peroxidase-labeled goat anti-rabbit secondary antibodies (Invitrogen, Carlsbad, USA) diluted at 1:500 in the blocking solution for 90 min in RT, washed in PBS-Tw (three times for 10 min at RT), in 0.05 M Tris-HCl pH 7.5 (for 10 min at RT), and stained in 10% 3,3’ diaminobenzidine in 0.05 M Tris-HCl pH 7.5 with 0.005% H_2_O_2_ for 10 min at RT. The reaction was stopped by rinsing in distilled water, dehydrating, and mounting in DPX mounting medium (Fluka, Buchs, Switzerland). The samples were examined under a BX51 microscope equipped with a DP80 CCD camera and cellSens software (Ver. 5174) (Olympus, Tokyo, Japan), and the images were reconstructed by stitching several Z stack series. The further 3D image analyses were performed in the FIJI ImageJ software (ImageJ 1.54f; http://imagej.org) [120] using the Iterative Deconvolution 3D plugin.

For fluorescence staining, the samples treated with primary antibody and washed in PBS-Tw were incubated with goat anti-rabbit IgG conjugated Alexa Fluor 488 (Life Technologies, Carlsbad, CA, USA) diluted 1:500 in the blocking solution for 90 min in RT, followed by rinsing with PBS-Tw (3 times for 10 min at RT in the dark). The samples were dehydrated and mounted in DPX mounting medium (Fluka, Buchs, Switzerland). The fluorescence signal was examined under the laser scanning confocal microscope FLUOVIEW FV3000 (Olympus, Tokyo, Japan) using the IMARIS 10.1 software (Oxford Instrument, Abingdon, UK) for 3D reconstruction of the Z-stack series. Examining the autofluorescence of human brain autopsy samples, enzymatic detection of bound primary antibodies was preferentially used for the detection of *Borrelia* in these samples.

The specificity of the primary antibodies to recognize both spiral and atypical forms of *B. burgdorferi* was verified by the application of antibodies to the paraplast sections of antibiotic-treated *B. burgdorferi* cultures mounted in agar (Appendix A).

## Figures and Tables

**Figure 1 ijms-24-16906-f001:**
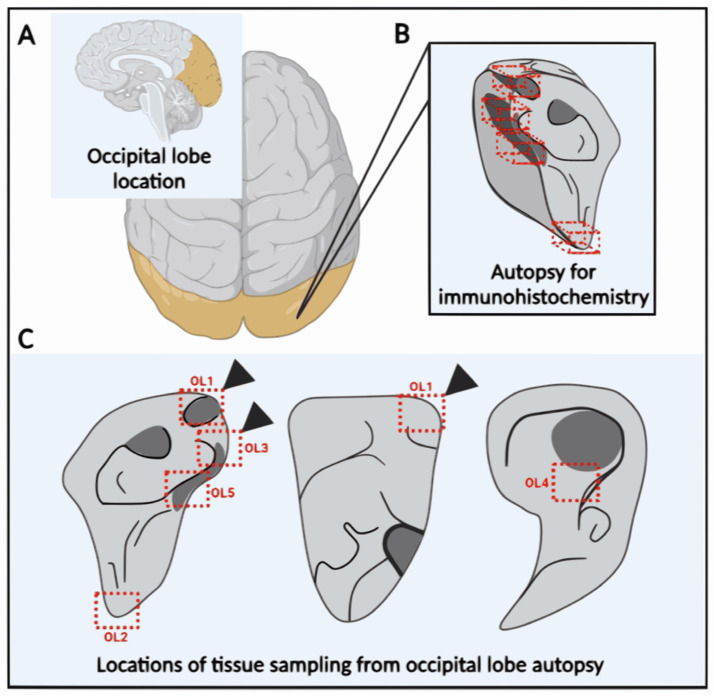
The scheme of the occipital lobe autopsy includes a description of sampling for immunohistochemistry detection. (**A**) Location of the occipital lobe (yellow) in the human brain. (**B**) The location of the collected autopsy samples (6 × 4 × 2 cm, approximately). (**C**) The red boxes and cubes indicate the five screening sites (each about 125 mm^3^) used for the immunohistochemical investigation. Arrow heads point to the locations where a positive signal was found.

**Figure 2 ijms-24-16906-f002:**
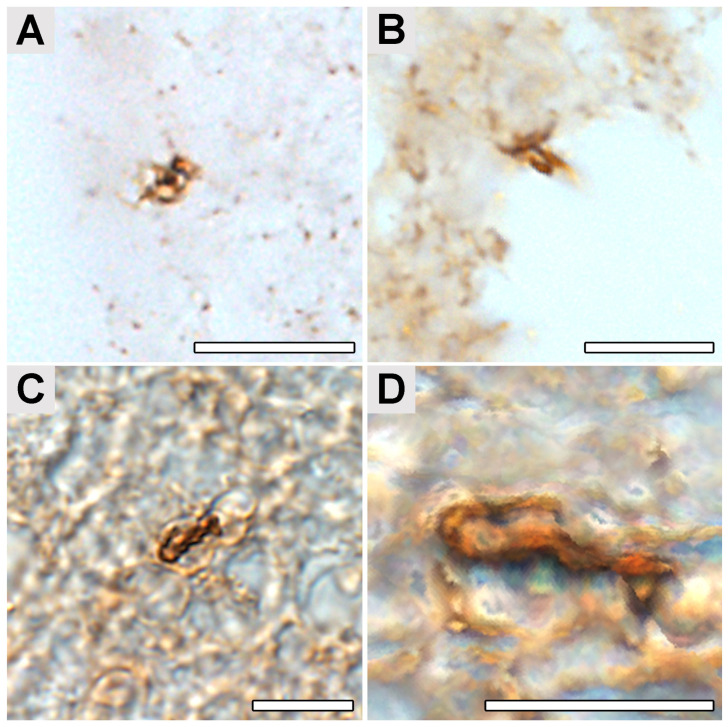
Atypical *Borrelia*-like structures in the paraplast section of a sample from the patient’s occipital lobe. Anti-*B. burgdorferi* polyclonal antibodies visualized by horseradish peroxidase-conjugated secondary antibodies were used for the detection. (**A**–**C**) Images from a light microscope show structures resembling atypical forms of *Borrelia*. (**D**) The image “(**C**)”was edited by image analysis using the FIJI ImageJ software (ImageJ 1.54f; http://imagej.org). Scale bar: 10 μm.

**Figure 3 ijms-24-16906-f003:**
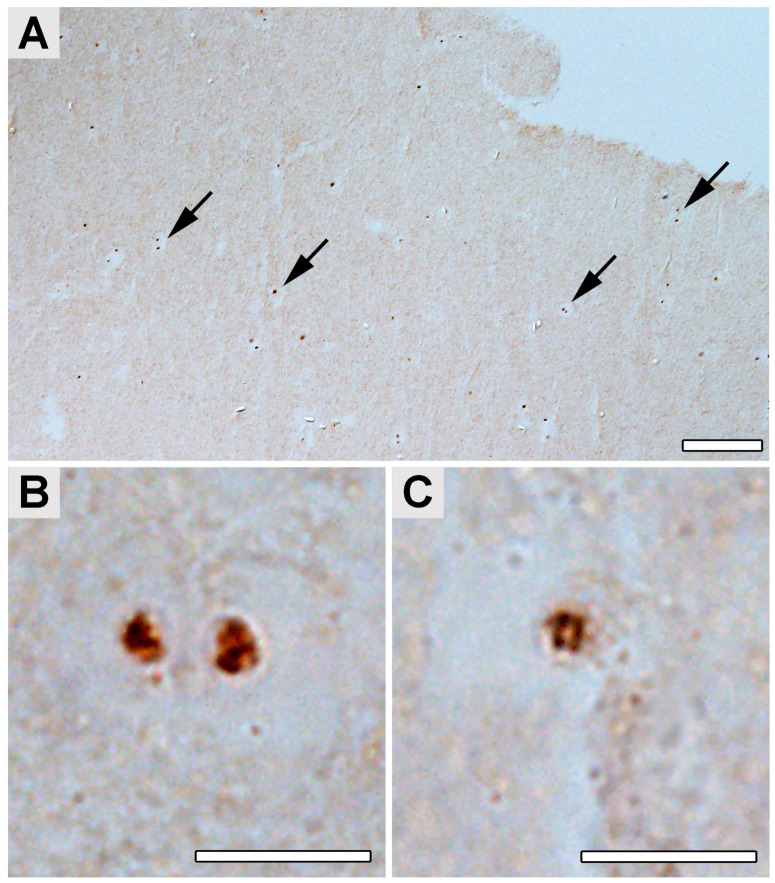
*Borrelia* cyst-like structures in the paraplast section of a sample from the patient’s occipital lobe. Anti-*B. burgdorferi* polyclonal antibodies visualized by horseradish peroxidase-conjugated secondary antibodies were used for the detection. The 3,3′ diaminobenzidine was used as a substrate for horseradish peroxidase. (**A**) Arrows pointing to the brown spots resemble *Borrelia* cyst-like structures. (**B**,**C**) High magnification of the *Borrelia* cyst-like structures. Scale bar: (**A**) 100 μm; (**B**,**C**) 10 μm.

**Figure 4 ijms-24-16906-f004:**
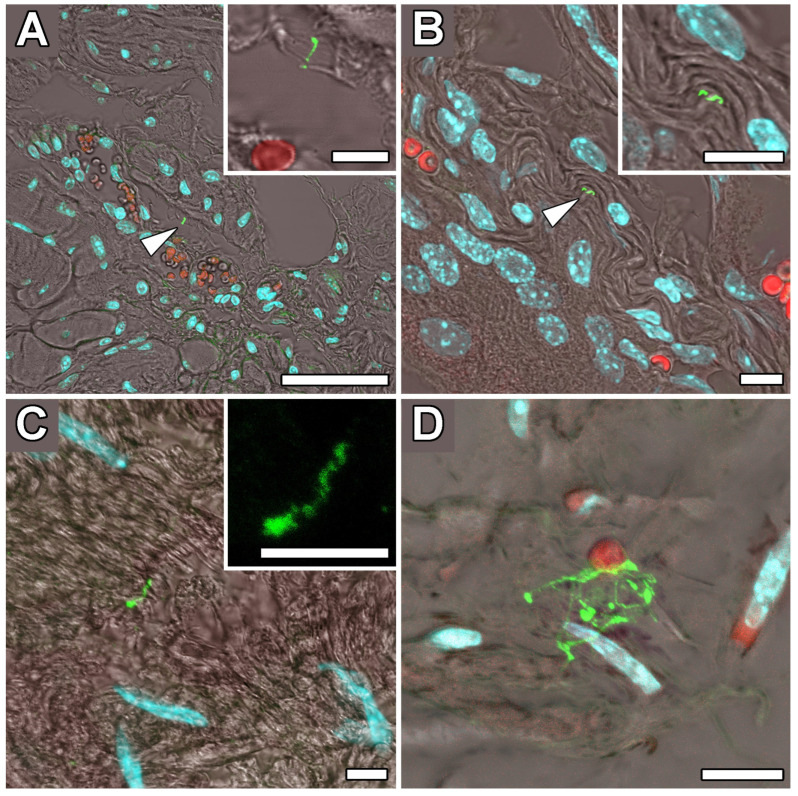
Laser scanning confocal images of persistent forms of *B. burgdorferi* s.s. in paraplast tissue sections of the C3H/HeN mouse treated with doxycycline and amoxicillin after establishment of infection. (**A**,**B**) Persistent form of *Borrelia* in the bladder (arrowheads). (**C**,**D**) Persistent form of *Borrelia* in the knee joint. Insets show close-up views of atypical forms of *Borrelia*. *Borrelia* was detected by *B. burgdorferi* polyclonal antibodies and visualized by fluorescently labeled (Alexa Fluor 488) secondary antibodies (in green), nuclei stained by DAPI (in blue), and autofluorescence of erythrocytes (in red). Scale bar: (**A**) 100 μm, (**B**–**D**) and insets 10 μm.

**Table 1 ijms-24-16906-t001:** Results of PCR amplification of *Borrelia* genes in different brain loci.

Gene Sample	*fla B* 388 bp	*ospC* 617 bp	*clpA* 706 bp	*clpX* 721 bp	*nifS* 629 bp	*rplB* 720 bp	*pepX* 666 bp	*pyrG* 687 bp	*uvrA* 677 bp
1—temporal lobe	*Bb* s.s.	*Bb* s.s.	+		+	+	+		+
2—choroid plexus	*Bb* s.s.		+		+	+	+	+	+
3—occipital lobe		*Bb* s.s.	+		+	+	+	+	+
4—frontal lobe	*Bb* s.s.	*Bb* s.s.	+		+	+	+	+	+
5—parietal lobe	*Bb* s.s.	*Bb* s.s.	+	+	+	+	+	+	+
6—basal ganglia	*B. gar*	*B. gar*			+	+	+	+	+
7—cerebellum	*B. gar*					+	+	+	+

## Data Availability

All data obtained during our study and supporting reported results can be found in this manuscript or Appendix A.

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
