# Peer review of "Concurrent Infection of the Human Brain with Multiple Borrelia Species"

_ijms, 2023, doi:10.3390/ijms242316906_

Round 1
Reviewer 1 Report
Comments and Suggestions for Authors
General Comments
The manuscript by Golovchenko and colleagues entitled “Concurrent infection of the human brain with multiple species of Lyme disease spirochetes” is an autopsy case report of a patient with Lyme neuroborreliosis (LNB) who was found by PCR to have both B. burgdorferi sensu stricto (Bbss) and B. garinii (Bg) in brain tissue. The first author and senior author have published extensively on tick-borne diseases, and they have made signifiant contributions to our understanding of these illnesses. The current manuscript adds to their achievements by demonstrating two species of Borrelia in different areas of the brain for the first time, and the provocative findings merit publication.
Major Comments
1. Line 127: According to the text, the patient was first diagnosed with Lyme disease in 2004 at age 8. Is this correct or should it be 2014? It appears that the patient never received IV antibiotics for LNB. This would be unusual for the time frame described in the case report and could account for the poor clinical outcome.
2. Lines 253 and 532: Polyclonal antibodies were used for immunostaining of brain tissue. Although the source for these antibodies is provided, more details about the binding specificity of the antibodies should be given with a published reference. Why were monoclonal antibodies not used?
3. Table 1 shows that distinct species of Borrelia were found in different portions of the brain in this case. A recent brain imaging study found that Borrelia-related changes were present in white matter as well as grey matter in cases of “post-treatment Lyme disease” (Marvel et al, PLoS ONE 17(10):e0271425). Could this tissue distinction be made in the autopsy case?
4. The findings raise many questions about how the patient acquired two species of Borrelia, and further discussion of this issue would be useful. Multiple species of Borrelia are found in ticks in various regions (Cakic et al, Med Vet Entomol. 2019;33:512-520; Cutler et al, Ticks Tick Borne Dis. 2021;12:101607; Gomez-Chamorro et al, Curr Res Parasitol Vector Borne Dis. 2021;1:100049.), and apparently little is known about coinfection of ticks with these Borrelia pathogens. The risk of single-exposure transmission of multiple Borrelia species versus multiple exposures should be discussed.
5. Aside from point #4, the discussion of Lyme symptomatology and novel antimicrobial treatments is too long, redundant and somewhat irrelevant for this autopsy case report, and much of the section from lines 350-400 could be eliminated.
Minor Comments
1. Line 108: Define “ATB” abbreviation
2. Lines 379-381: Sentence is ungrammatical and should be eliminated.
3. Line 20 (Supplementary Material): “Immunodetection of Borrelia on paraplast sections of cultured spirochetes was performed using the same protocol as for human brain tissue (above).” There is no above.
Comments on the Quality of English LanguageGood
Author Response
Authors response to Reviewer 1.
We are grateful for all the help that led to improvement of our manuscript.
We addressed all your questions and comments appropriately. Thank you!
Major Comments
- Line 127: According to the text, the patient was first diagnosed with Lyme disease in 2004 at age 8. Is this correct or should it be 2014? It appears that the patient never received IV antibiotics for LNB. This would be unusual for the time frame described in the case report and could account for the poor clinical outcome.
According to the information that the authors obtained in regards to analyzed case, the data presented in the manuscript are correct. The first diagnose of Lyme disease was made in 2004. Even though the patient underwent multiple antibiotic treatments during the course of his disease, unfortunately there was no records of IV ATB therapy. All medications were taken by patient perorally, as mentioned in the manuscript.
- Lines 253 and 532: Polyclonal antibodies were used for immunostaining of brain tissue. Although the source for these antibodies is provided, more details about the binding specificity of the antibodies should be given with a published reference. Why were monoclonal antibodies not used?
A commercially prepared rabbit polyclonal antibody used to detect Borrelia was directed against several different proteins specific to the B. burgdorferi sensu lato complex (Borrelia burgdorferi Polyclonal Antibody, Invitrogen, catalogue number PA1-73004). It recognizes a membrane vesical protein of 83 kDa, a flagellar protein, FlagB, of 41 kDa and the specific membrane proteins, OspA and OspB, with molecular masses of 34 kDa and 31 kDa. The PA1-73004 antibody was raised against B. burgdorferi whole cell preparation. Product specific informed that PA1-73004 has been successfully used in Western blot, Immunohistochemistry (paraffin), and immunofluorescence applications. By Western blot, this antibody detects bands at 83, 41, 34 & 31 kDa. This antibody has been used previously to specifically mark B. burgdorferi in monolayers of endothelial cells (Yuste et al., 2022;Sapiro et al., 2023). In addition, we also tested the specificity of the mentioned antibody on B. burgdorferi tissue culture.
We consider the detailed description of the above mentioned polyclonal antibodies in the manuscript unnecessary, as they are widely used and are the part of common laboratory reagents in Borrelia-related research.
The reason for not using the monoclonal antibodies was added to the text of the manuscript: ” The polyclonal antibodies were selected to cover a broader spectrum of Borrelia antigens for possible binding, instead of monoclonal due of the origin of tested samples. Postmortem changes might limit the number of borrelial antigens for localization as not all antigens might be well preserved”.
- Table 1 shows that distinct species of Borrelia were found in different portions of the brain in this case. A recent brain imaging study found that Borrelia-related changes were present in white matter as well as grey matter in cases of “post-treatment Lyme disease” (Marvel et al, PLoS ONE 17(10):e0271425). Could this tissue distinction be made in the autopsy case?
Congestion and swelling were noted on macroscopic as well as histological examination. As this condition typically occurs as a result of death by hanging/suffocation, it is not possible to discern how much the finding is due to the manner of death and how much is due to another cause. White and gray matter had the usual histological structure.
- The findings raise many questions about how the patient acquired two species of Borrelia, and further discussion of this issue would be useful. Multiple species of Borrelia are found in ticks in various regions (Cakic et al, Med Vet Entomol. 2019;33:512-520; Cutler et al, Ticks Tick Borne Dis. 2021;12:101607; Gomez-Chamorro et al, Curr Res Parasitol Vector Borne Dis. 2021;1:100049.), and apparently little is known about coinfection of ticks with these Borrelia pathogens. The risk of single-exposure transmission of multiple Borrelia species versus multiple exposures should be discussed.
Czech republic belong to the countries where the Lyme disease is endemic with multiple spirochete species pathogenic for human widely represented in local tick vectors. The presence of multiple pathogen species in local Ixodes ricinus ticks (cases of co-infections) are well known in this region and the data are widely discussed in ecological publications. Citations of such papers were added.
- Aside from point #4, the discussion of Lyme symptomatology and novel antimicrobial treatments is too long, redundant and somewhat irrelevant for this autopsy case report, and much of the section from lines 350-400 could be eliminated.
DONE
Minor Comments
- Line 108: Define “ATB” abbreviation - DONE.
- Lines 379-381: Sentence is ungrammatical and should be eliminated. - DONE
- Line 20 (Supplementary Material): “Immunodetection of Borrelia on paraplast sections of cultured spirochetes was performed using the same protocol as for human brain tissue (above).” There is no above.
CORRECTED!
Reviewer 2 Report
Comments and Suggestions for Authors
I don't think that in the presented case, there is adequate substantiation for the causal association between borrelial infection and the patient's clinical symptoms and I do not feel competent to give a review for this manuscript. I propose the microbiologic evidence/results to be reviewed by microbiologists and provide some comments in the attached file.

Author Response
The Authors greatly appreciate all the help, advices and recommendations that, no doubts, improved the quality of our manuscript. Comments are marked in attached file.

Reviewer 3 Report
Comments and Suggestions for Authors
The manuscript deals with mixed infection with Borrelia spp. It describes a case case report of the detection of spirochetal DNA in multiple loci of the brain of a Lyme disease patient. Considering the circumstances, revision of the manuscript as a Case Report should be considered. Taking this into consideration, a slight text reduction should be made.
Title – please remove dot and adapt to read as: Concurrent infection of the human brain with multiple Borrelia spp.
Abstract and main text – write Latin names in italics; draw text in a single paragraph
Line 30 – replace that with which
Keywords – display alphabetically
Introduction – write Latin names in italics
Line 39 – remove (s. l.)
Line 42 – write each and every species name the first time it is presented – Borrelia afzelii
Line 46 – replace pathologies with pathology
Line 56 – 6 months
Line 74 – B. burgdorferi s. l.
Line 76 – B. afzelii; replace that with which
Line 87 – replace brains with brain
Line 91 – remove (s.s.)
Line 104 – please better describe that xenodiagnostic study – which vector was it, etc.?
Line 108 – describe the meaning of ATB
Line 111 – define IHC
Line 129 – are unavailable (instead of is unavailable)
Line 133 – anti-Borrelia
Line 136 – (olanzapin)… (vortioxetin) – indicate manufacturers for the trademarks
Line 146 – Bartonella quintana … Babesia spp. (Babesia microti and Babesia divergens)
Line 148 – write antibiotic names with lowercase initials – mention manufacturer(s)
Lines 298, 310, 311, etc. – abbreviate species names
Line 315 – Borrelia spp.
Line 335 – “Trojan horse”
Line 356 – explain the meaning of CLD
Comments on the Quality of English LanguageMinor editing of English language required.
Author Response
Authors response to Reviewer 3
We would like to thank you for your time and all your help that lead to improvement of our manuscript. We addressed all your comments appropriately. THANK YOU SO MUCH!
Considering the circumstances, revision of the manuscript as a Case Report should be considered. Taking this into consideration, a slight text reduction should be made.
DONE
Title – please remove dot and adapt to read as: Concurrent infection of the human brain with multiple Borrelia spp.
DONE
Abstract and main text – write Latin names in italics; draw text in a single paragraph - DONE
Line 30 – replace that with which - DONE
Keywords – display alphabetically- DONE
Introduction – write Latin names in italics- DONE
Line 39 – remove (s. l.) - DONE
Line 42 – write each and every species name the first time it is presented – Borrelia afzelii- DONE
Line 46 – replace pathologies with pathology- DONE
Line 56 – 6 months- DONE
Line 74 – B. burgdorferi s. l. - DONE
Line 76 – B. afzelii; replace that with which- DONE
Line 87 – replace brains with brain- DONE
Line 91 – remove (s.s.) - DONE
Line 104 – please better describe that xenodiagnostic study – which vector was it, etc.?- Description was extended. DONE
Line 108 – describe the meaning of ATB- DONE
Line 111 – define IHC- DONE
Line 129 – are unavailable (instead of is unavailable) - DONE
Line 133 – anti-Borrelia- DONE
Line 136 – (olanzapin)… (vortioxetin) – indicate manufacturers for the trademarks- DONE
Line 146 – Bartonella quintana … Babesia spp. (Babesia microti and Babesia divergens) - DONE
Line 148 – write antibiotic names with lowercase initials – mention manufacturer(s) - DONE
Lines 298, 310, 311, etc. – abbreviate species names- DONE
Line 315 – Borrelia spp. - DONE
Line 335 – “Trojan horse” - DONE
Line 356 – explain the meaning of CLD- DONE
Round 2
Reviewer 1 Report
Comments and Suggestions for Authors
Line 652: With regard to the Invitrogen antibody, although a detailed description is not needed, the Invitrogen catalogue number (PA1-73004) and relevant references (Yuste et al., 2022; Sapiro et al., 2023) should be included in the manuscript.
Line 546: "(cca. 4 weeks)" can be omitted.
Comments on the Quality of English Language
Line 341: Avoid colloquial contractions ("doesn't")
Author Response
RESPONSE to REVIEWER 1 (Minor revision)
We are grateful for all the help of Reviewer 1 in improving of our manuscript. All comments and advices were addressed in the second revision. Thank you!
Comments and Suggestions for Authors
Line 652: With regard to the Invitrogen antibody, although a detailed description is not needed, the Invitrogen catalogue number (PA1-73004) and relevant references (Yuste et al., 2022; Sapiro et al., 2023) should be included in the manuscript.
The catalogue number of polyclonal antibodies was added to the text.
Recommended relevant references of Yuste et all., 2022 and Sapiro et al., 2023 were included in manuscript.
Yuste RA, Muenkel M, Axarlis K, Gómez Benito MJ, Reuss A, Blacker G, Tal MC, Kraiczy P, Bastounis EE Borrelia burgdorferi modulates the physical forces and immunity signaling in endothelial cells. iScience. 2022 Jul 20;25(8):104793. doi: 10.1016/j.isci.2022.104793
Sapiro AL, Hayes BM, Volk RF, Zhang JY, Brooks DM, Martyn C, Radkov A, Zhao Z, Kinnersley M, Secor PR, Zaro BW, Chou S Longitudinal map of transcriptome changes in the Lyme pathogen Borrelia burgdorferi during tick-borne transmission. Elife 2023 Jul 14:12:RP86636. doi: 10.7554/eLife.86636.
Line 546: "(cca. 4 weeks)" can be omitted.
(cca.4 weeks) remark was deleted from the text.
Line 341: Avoid colloquial contractions ("doesn't")
Correction was done: “doesn’t” was substituted by “does not”.
Comments on the Quality of English Language: Minor editing of English language required
The re-submitted version of revised manuscript was extencively checked and English language was edited by Professor of Biology Vett Lloyd from Mt. Alison University (New Brunswick, Canada).
Reviewer 2 Report
Comments and Suggestions for Authors
In this paper, the authors provide a comprehensive description of the atypical LB presentations and antibiotic refractory cases and the possible pathophysiologic mechanisms for it. However, my main objection as provided in the previous review remains the same: the crucial information for establishing a diagnosis of Lyme neuroborreliosis is missing in the presented case and this should have been clearly emphasised.
"The information on results of the biochemical analysis of cerebrospinal fluid including indexes of intrathecal synthesis of borrelial antibodies is missing. This is essential for the diagnosis of Lyme neuroborreliosis.
In the presented case, there is inadequate substantiation for the causal association between borrelial infection and clinical symptoms in the first place. On the contrary, lack of efficacy of antibiotics in this case favors the assumption that borrelial infection might not have been the cause of symptoms."
Author Response
Response to Reviewer 2 comments
We are grateful to Reviewer for the time and helpful discussion that led to improvement of our manuscript. THANK YOU!
In this paper, the authors provide a comprehensive description of the atypical LB presentations and antibiotic refractory cases and the possible pathophysiologic mechanisms for it. However, my main objection as provided in the previous review remains the same: the crucial information for establishing a diagnosis of Lyme neuroborreliosis is missing in the presented case and this should have been clearly emphasized.
"The information on results of the biochemical analysis of cerebrospinal fluid including indexes of intrathecal synthesis of borrelial antibodies is missing. This is essential for the diagnosis of Lyme neuroborreliosis.
In the presented case, there is inadequate substantiation for the causal association between borrelial infection and clinical symptoms in the first place. On the contrary, lack of efficacy of antibiotics in this case favors the assumption that borrelial infection might not have been the cause of symptoms."
Response of Authors:
To the Reviewers’ objection, we would like to state that not a single time through the text of the manuscript we did declare or stated that the patient was suffering from neuroborreliosis, particularly because of the absence of positive proof of intrathecal Borrelia-specific antibodies synthesis. Specific antibodies against Borrelia were determined post mortem in both serum and cerebrospinal fluid with similar results. Because of the manner of death, blood contamination was macroscopically visible in the cerebrospinal fluid, and thus the index of intrathecal synthesis of Borrelia-specific immunoglobulins could not be reliably calculated.
Due to the methodology based on the post-mortem analysis of brain tissue, our study is not able unambiguously confirm the connection between the patient's subjective complaints and the active Borrelia infection. And, in the presented manuscript, the authors never insisted on such statement. The possible pathological effect of persistent antigenic structures of Borrelia detected in present study is also discussed from the point of view of the potential induction of an immunopathological reaction, associated with the chronic inflammation and/or autoreactive immune response, where non-replicative structures can also be recognized as long-term antigenic stimuli (https://www.mdpi.com/2075-1729/13/2/527). Further experimental work is needed to elucidate this possible pathological mechanism that would help explain the persistence of symptoms despite intensive antibiotic treatment.
Round 3
Reviewer 2 Report
Comments and Suggestions for Authors
I think that the autors' statement provided in their reply "not a single time through the text of the manuscript we did declare or stated that the patient was suffering from neuroborreliosis" is not aligned well with the authors' statement "in this LD patient's brain" already presented in the abstract. Again, I think that a strict destinction between detecting borrelial DNA/antigens in the brain tissue of a deceased young man in whom LD/Lyme neuroborreliosis has not been confirmed while he was alive and LD/Lyme neuroborreliosis should have been made in order to avoid possible misleading interpretation of such findings. I have no further comments.